# Genome mining and characterisation of biosynthetic clusters in *Aspergillus japonicus* isolated from the Amazon rainforest

Josy Caldas Rodrigues[1/+], Maria Eduarda Grisolia[1], Alice da Silva Queiroz[1],
Ana Luísa Rodrigues Lima[2], Clarice Virginia Santos Goiabeira[1], Leila de Mendonça Lima[3],
Ormezinda Celeste Cristo Fernandes[1]

[1]Fundação Oswaldo Cruz-Fiocruz, Instituto Leônidas & Maria Deane, Manaus, AM, Brasil
[2]Universidade Federal do Amazonas, Manaus, AM, Brasil
[3]Fundação Oswaldo Cruz-Fiocruz, Instituto Oswaldo Cruz, Rio de Janeiro, RJ, Brasil

**BACKGROUND** In light of the biotechnological potential demonstrated by *Aspergillus* species and, considering the great need for further research into the search for new sources of active molecules and the biodiversity of these microorganisms occurring in the Amazon region.

**OBJECTIVES** This research aimed to investigate the biotechnological potential of the fungus *Aspergillus japonicus* Amazon Fungi Collection (CFAM) 0234, a fungal strain isolated from Amazonian soil and stored in the CFAM.

**METHODS** For this purpose, the *Aspergillus* species was investigated through comparative genomic analysis and antimicrobial activity assays.

**FINDINGS** Genome sequencing revealed a fragmented assembly (72.67 Mbp, N50 = 152 kbp) containing 106 biosynthetic clusters (BGCs), surpassing the reference strain CBS 114.51 (57 BGCs). Among the clusters identified, NRPS, PKS type I and hybrid NRPS-PKS systems stood out, including clusters exclusive to betalactones and isocyanides, potentially involved in the synthesis of β-lactam antibiotics and innovative metabolites. BiG-SCAPE analysis identified 63 BGC families unique to CFAM 0234, suggesting evolutionary adaptations to the competitive environment of the Amazon. Biological assays demonstrated selective antimicrobial activity of the ethyl acetate extract against *Escherichia coli*, *Shigella sonnei* and *Sthapylococcus aureus* (MRSA), with inhibition halos ranging from 8 mm to 6 mm in diameter, pathogens classified as priorities for research into new antibiotics. The correlation between predicted BGCs and antimicrobial activity reinforces the strain's biotechnological potential. Despite the fragmentation of the genome, the high completeness assessed by BUSCO (98.5%) confirms the quality of the assembly, while the detection of single nucleotide polymorphisms (SNPs) in regulatory regions and rearrangements close to BGCs suggests evolutionary pressure for metabolic diversification. The lack of correspondence with the minimum information about a biosynthetic gene cluster (MIBiG) bank and the limitations of crude extracts highlight the need for complementary techniques, such as long-read sequencing (Oxford Nanopore) and metabolomic analysis [liquid chromatography-mass espectrometry (LC-MS)], to link clusters to active metabolites.

**MAIN CONCLUSIONS** *Aspergillus japonicus* CFAM 0234 represents a promising microbial resource for bioprospecting in the Amazon, offering relevant genomic and chemical insights for the development of new antimicrobial agents. Future studies will focus on the purification of compounds and activation of silent BGCs, aiming at sustainable pharmaceutical applications.

Key words: bioprospecting - Amazon - *Aspergillus japonicas* - biosynthetic clusters - antimicrobial resistance

One of the great desires related to the Amazon region is associated with the possibility of curing various diseases through the development of medicines sourced from the region's biodiversity.[1]

In this sense, studies involving the bioprospecting of fungi of the genus *Aspergillus* from Amazonian soil have shown that these microorganisms are an alternative source in the search for new active molecules capable of inhibiting microorganisms such as *Candida albicans*, *Escherichia coli* and *Staphylococcus aureus* that are recognised for causing hospital infections and presenting resistance to available drugs.[2,3,4]

However, elucidating the secondary metabolites responsible for these desired activities is still a challenge - many studies are carried out using organic extracts and fractions due to the financial and technical difficulties involved in isolating and identifying these metabolites, thus hampering the development of innovative products.[5]

Financial support: FAPEAM (PRODOC - Post-Doctorate Support Program), ILMD/FIOCRUZ.
CR and MEG contributed equally to this work.
+ Corresponding author: josy.silva@fiocruz.br | ● https://orcid.org/0000-0002-7886-7929

**Handling editor:** Adeilton Alves Brandão | ● https://orcid.org/0000-0001-5877-607X

Sequencing combined with genome mining using bioinformatics algorithms enables the identification of groups of secondary metabolism genes. As an example, citric acid fermentation by *Aspergillus niger* has been studied for almost 100 years, but the elucidation of the central carbon metabolism involved in the process was only possible with the availability of genome data.[6]

Therefore, analysis of the genome of microorganisms provides a better understanding of their fundamental cellular processes and metabolic potential, contributing to solving urgent and important challenges in health, enzyme biotechnology, bioenergy and ecological diversity. The availability of whole genomes allows the search for bioactives with enhanced properties and provides invaluable help to improve the production of biotechnological products.[7]

Based on the above considerations and the great need for new sources of active molecules, the aim of this work was to investigate the biotechnological potential, based on comparative genomic analysis, of the fungus *A. japonicus* stored in the Amazon Fungi Collection (CFAM).

## MATERIALS AND METHODS

*Acquisition of strains, DNA extraction and sequencing* - The fungal strain used in this study was selected at the Multiuser Health/Mycology Laboratory of the Leônidas and Maria Deane Institute (ILMD/FIOCRUZ Amazônia). Genomic sequencing was conducted at the Genomics Technology Platform, linked to the Technological Development Program in Health Supplies (PD-TIS), under the coordination of the Vice-presidency of Research and Reference Laboratories at FIOCRUZ.

The microorganisms used as tests were supplied by the Amazon Bacteria Collection (CBAM), the CFAM and the INPA Collection of Microorganisms of Medical Interest. For this study, the *A. japonicus* strain (CFAM 0234), isolated from the soil of the Amazon biome and stored in the CFAM-ILMD (FIOCRUZ), was selected. The strain was reactivated in Petri dishes containing malt extract agar (MEA) culture medium and incubated at 28ºC for seven days in a BOD incubator.[8]

Genomic DNA was extracted using the DNeasy Blood and Tissue kit (Qiagen), which is widely used to obtain high-quality DNA from fungi. The quality of the extracted DNA was assessed using 1.0% agarose gel electrophoresis, stained with GelRed, and quantification was carried out using spectrophotometry. The genomic DNA was prepared and sequenced on the Next Generation Nucleic Acid Sequencing Platform (FIOCRUZ/IOC - RJ).

The genomic libraries were prepared using the Illumina DNA Prep protocol, with a fragmentation time of 15 to 20 minutes at 55ºC, in order to optimise the yield due to the characteristics of the fungal DNA.

Sequencing was carried out on the Illumina NextSeq2000 platform using the P3 300-cycle kit, which generates approximately 1.2 billion reads per run. A negative control ($H_2O$) was included to monitor the quality of the library construction process.

*Genome assembly, quality assessment and annotation* - Raw reads were assessed for quality with FastQC v0.11.8 and processed with Trimmomatic (parameters: SLIDINGWINDOW:4:20 MINLEN:50) to remove adapters, low-quality sequences and truncated reads.[9] The assembly was carried out with SPAdes v3.13,[10] using a hybrid strategy that combined *trusted* contigs, based on the incomplete genome of *A. japonicus* CBS 114.51 (scaffold level, N50 = 412 kbp; NCBI accession: GCA_003184785.1), and *untrusted* contigs, generated by an initial *novel* assembly with corrected reads. The quality of the assembly was assessed with QUAST v5.0.2,[11] for analysis of fragmentation metrics (N50, number of contigs), and compared with other *A. japonicus* strains (strains Y4009A, accession GCA_023625335.1, and PSFR, accession GCA_016808025.1). Genomic completeness was checked with BUSCO v5.4.3,[12] using the aspergillus_odb12 database.

To identify orthologous genes, the analysis was conducted with OrthoFinder v2.5.4,[13] using predicted proteomes from *A. japonicus* CFAM 0234 (annotated with Funannotate v1.8.1) and CBS 114.51 (NCBI public annotation). The aim was to identify groups of conserved and unique genes in order to explore functional and evolutionary differences between the strains.

*Analysis of biosynthetic clusters (BGCs) and genomic comparison* - The prediction of BGCs was carried out with antiSMASH v7.0,[14] using the detection *strictness* parameter to identify enzyme systems associated with the production of secondary metabolites, including polyketide synthases (T1PKS, T3PKS), non-ribosomal peptides (NRPS) and hybrid systems (NRPS-PKS, terpenes, siderophores). To group the BGCs into gene cluster families (GCFs) and compare profiles between the strains, the GenBank files of the predicted BGCs (CFAM0234 and CBS 114.51) were analysed with BiG-SCAPE v1.1.5,[15] the process is now being scaled up to mine entire genera, strain collections and microbiomes. However, no bioinformatic framework is currently available for effectively analyzing datasets of this size and complexity. In the present study, a streamlined computational workflow is provided, consisting of two new software tools: the 'biosynthetic gene similarity clustering and prospecting engine' (BiG-SCAPE using a similarity threshold of 30% (default value for conservative clustering) and Jaccard's distance to calculate similarity.

The genomic alignment between *A. japonicus* CFAM 0234 and the reference strain CBS114.51 was performed with MUMmer4.[16] Parameters such as - *maxmatch* (for identifying all exact matches) and -*c 1000* (filtering out short alignments) were used. The raw results (delta files) were refined with the *delta-filter* tool, retaining only regions with identity ≥ 99% and length ≥ 500 kb. The coordinates of the alignments were extracted with *show-coords*, generating a tabular file with genomic positions, identity and orientation of the sequences. To detect single nucleotide polymorphisms (SNPs), *show-snps* was used with the -*C* parameter (exclusion of repetitive regions), followed by manual filtering to remove variants in regions of low complexity or close to gaps.

The distribution of biosynthetic clusters was analysed quantitatively, comparing the relative abundance of each class among the strains. For integrated visual-

isation, BGCs, SNPs and structural rearrangements were mapped on the genome with Circos v0.69-9.[17] The alignment data was processed in R, converting coordinates to the appropriate format for the Circos *karyotype*, and plotted in circular graphs highlighting the location of regions of metabolic interest and genomic variations.

*Experimental validation: biological tests* - To confirm the production of the metabolites with antimicrobial action, biological activity tests were carried out.

*Extraction of secondary metabolites* - The fungal culture was grown in 500 mL Erlenmeyer flasks containing 200 mL of YES culture medium [Czapeck yeast extract 2% (w/v) and sucrose 20% (w/v)], incubated at 28ºC for seven days. After growth, the secondary metabolites were cold extracted with 200 mL of P.A. grade ethyl acetate (AcOEt) (SYNTH). The AcOEt extract was kept in contact with the culture for 48 h, filtered on Whatman No. 30 paper and concentrated in an exhaust hood (SCILOGEX RE 100-Pro) to assess antimicrobial activity.[18]

*Confirmation of antibacterial activity* - The antibacterial activity of the extract of *A. japonicus* CFAM 0234 was evaluated by the agar diffusion technique per well, using pathogenic bacteria of clinical origin frequently associated with hospital infections (Table I). The bacterial strains selected included *Acinetobacter nosocomialis*, *Enterococcus faecium*, *E. coli*, *Klebsiella pneumoniae*, *S. aureus* (MRSA) and others (Table I).

The bacteria were seeded on Petri dishes (10 mm × 90 mm) containing Mueller-Hinton agar and incubated at 37ºC for 24 h. From the cultures, a cell suspension adjusted to turbidity equivalent to the McFarland 0.5 scale was prepared. Next, 150 µL of the suspension was sown evenly on the surface of Mueller-Hinton agar. Three 5 mm diameter wells were drilled in each plate, into which 150 µL of the fungal extract (1 mg/mL) were added, solubilised in sterile distilled water and 4% (v/v) dimethyl sulfoxide (DMSO). As controls, vancomycin (1 mg/mL; Merck, Germany) was used as a positive and a mixture of sterilised distilled water/DMSO 4% (v/v) as a negative. The plates were incubated at 37ºC for 24 h and antibacterial activity was determined by measuring the inhibition halo (in millimetres). All tests were carried out in triplicate.[18]

## RESULTS

*Genome sequencing* - Sequencing the genome of *A. japonicus* CFAM 0234 resulted in an assembly of 962 contigs, with a total size of 72.67 Mbp and an N50 value of 152 kbp. The GC content was 50.45%, and the largest contig reached 600 kbp. The *CBS 114.51* reference genome, available on NCBI, is also not complete, being at scaffold level, with a total size of 36.1 Mbp and an N50 of 412 kbp.

The comparative analysis with CBS 114.51 should be interpreted with caution, since both assemblies (CFAM 0234 and CBS 114.51) are fragmented. The apparent doubling of the size of the CFAM 0234 genome (72.67 Mbp vs. 36.1 Mbp) may reflect both biological differences and assembly artifacts, since regions missing from the reference could not be validated. On the other hand, BUSCO's assessment of genomic integrity indicated that the assembled genome shows a high level of completeness, with 4959 complete genes identified (339 single-copy and 4620 duplicated), as well as only 10 fragmented and 65 missing genes, out of a total of 5034 genes assessed.

Orthologue analysis using *OrthoFinder* compared the study strain with the CBS *114.51* reference genome. The total number of genes identified was 962 in the sequenced strain and 163 in the reference, with the majority grouped into orthologs (926 and 161, respectively).

TABLE I

Pathogenic bacteria of clinical origin used to evaluate antibacterial activity

| Bacteria/CBAM | Antibiotic resistance | Clinical origin |
|---|---|---|
| *Acinetobacter nosocomialis* 0725 | Cephalothin, oxacillin, penicillin and vancomycin | Oropharynx |
| *Enteroccocus faecium* 0639 | Amikacin, cephalothin, penicillin and oxacillin | Oropharynx |
| *Enteroccocus faecalis* 0735 | Cephalothin and oxacillin | Oral cavity |
| *Escherichia coli* 0001 | Ampicillin, cephalothin, oxocillin, penicillin, tetracillin and vancomycin | Diarrheal stools |
| *Escherichia coli* 0002 | Amikacin, ampicillin, cephalothin, cefotaxime, oxacillin, penicillin, piperacillin+tazobactam, tetracycline and vancomycin | Diarrheal stools |
| *Klebsiella aerogenes* 0717 | Amoxicillin, cephalothin, oxacillin, penicillin and vancomycin | |
| *Klebsiella pneumoniae* 0462 | Amoxicillin, oxacillin, penicillin and vancomycin | Oropharynx |
| *Proteus mirabilis* 0686 | Amikacin, cephalothin, oxacillin and vancomycin | Diarrheal stools |
| *Pseudomonas aeruginosa* 0024 | Amoxicillin, cephalothin, oxacillin and vancomycin | Diarrheal stools |
| *Salmonella entérica* 0047 | Oxacillin and vancomycin | |
| *Salmonella typhi* 0009 | Oxacillin and vancomycin | Skin lesion |
| *Shigella flexneri* 0046 | Amoxicillin, cephalothin, oxacillin and vancomycin | Diarrheal stools |
| *Shigella sonnei* 0036 | Amoxicillin, cephalothin, oxacillin and vancomycin | Diarrheal stools |
| *Sthapylococcus aureus* ATCC 4330 (MRSA) | Methicillin | - |

CBAM: Amazon Bacteria Collection.

Only 36 genes in the lineage studied and two in the reference were not assigned to orthogroups. Several contigs in the lineage analysed were associated with specific orthogroups, with genes distributed in contigs such as *Contig_591*, *Contig_606*, *Contig_681*, and others standing out, which may indicate regions of functional interest that have not yet been fully characterised.

*Comparative analysis of BGCs* - The CFAM 0234 strain revealed an expanded repertoire of BGCs, with 106 BGCs identified compared to 57 in the CBS 114.51 reference strain. BiG-SCAPE analysis using a 30% similarity threshold showed that 63 BGC families (GCFs) are unique to CFAM 0234, while 33 are unique to CBS 114.51 (Table II). The lack of correspondence with clusters from the minimum information about a biosynthetic

gene cluster (MIBiG) bank reinforces the unique biotechnological potential of these strains.

The comparative map generated by Circos (Figure A) highlights the association between structural rearrangements and the location of BGCs, particularly NRPS and T1PKS clusters. These regions showed the highest density of SNPs, suggesting evolutionary pressure for metabolic diversification. CFAM 0234 exhibits greater BGC complexity, with 34 NRPS clusters (vs. 15 in the reference) and 12 NRPS-T1PKS hybrid clusters (vs. 7), including unique systems such as betalactones and isocyanides (Figure B).

Among the 96 shared GCFs, clusters with conserved enzyme profiles stand out, such as FAM_00001 (T1PKS) and FAM_00009 (NRPS-T1PKS), which may represent essential metabolic pathways for the fungus' ecological niche (Table III). Exclusive GCFs, such as FAM_00096 (T1PKS-isocyanide) in CFAM 0234, are priority candidates for functional characterisation. Table III summarises the disparity in the distribution of BGC classes between the strains, highlighting exclusive clusters such as FAM_00096 (T1PKS + Isocyanide) in CFAM 0234 and shared ones such as FAM_00009 (NRPS + T1PKS), with potential for antifungal hybrid metabolites.

*Antibacterial activity of the CFAM 0234 strain extract* - The ethyl acetate extract (AcOEt) of CFAM 0234 selectively inhibited *E. coli* CBAM 0001, *Shigella sonnei* CBAM 0036 and *Sthapylococcus aureus* ATCC 4330 (MRSA), with inhibition halos of 8 mm, 8 mm and 6 mm in diameter, respectively (Table IV).

TABLE II

Overview of biosynthetic gene clusters families (BGCFs)

| Métric | CFAM 0234 | CBS 114.51 | Both |
|---|---|---|---|
| Total GCFs | 89 | 59 | 125 |
| Exclusive GCFs* | 63 (71%) | 33 (56%) | - |
| Shared GCFs** | - | - | 96 (77%) |
| Similarity to MIBiG | 0% | 0% | 0% |

*GCFs containing only BGCs from one strain; **families with BGCs from both strains (similarity > 30%); CFAM: Amazon Fungi Collection.

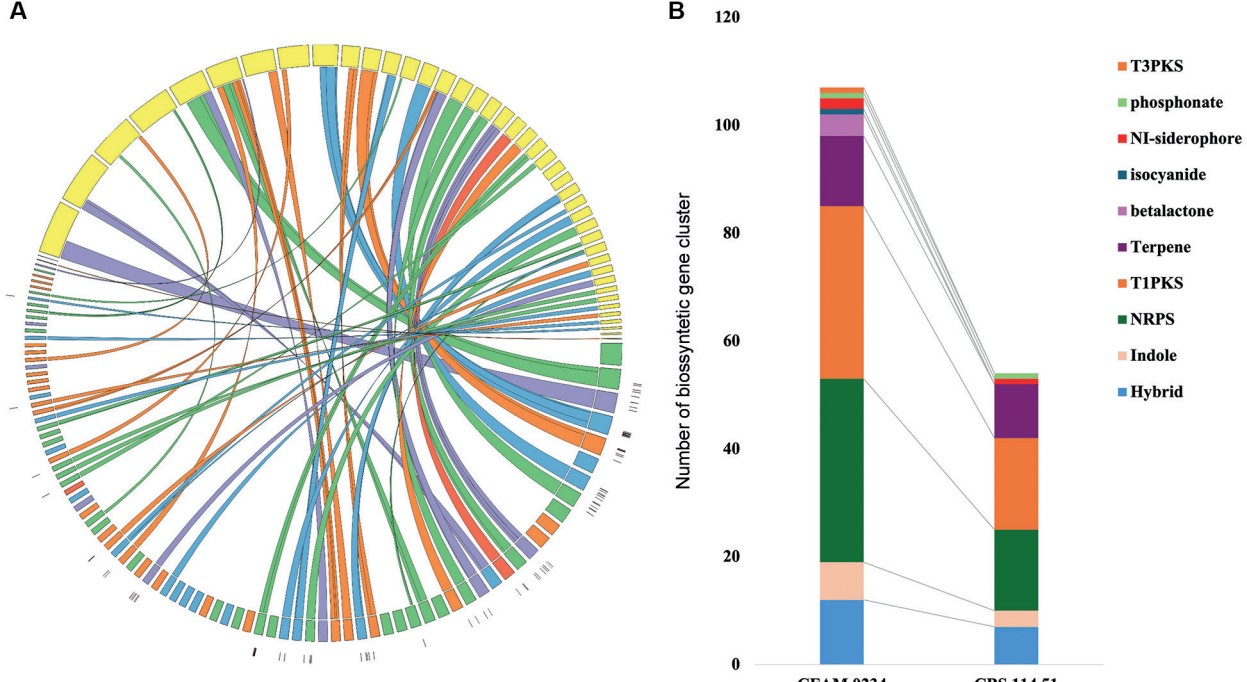

Comparative cluster analysis of biosynthesis genes in *Aspergillus japonicus*. (A) Circos representation of the alignment between strain Amazon Fungi Collection (CFAM) 0234 (coloured internal contigs) and reference strain CBS 114.51 (yellow external contigs). Colours in the CFAM 0234 contigs indicate biosynthetic clusters (BGCs) classes: green [non-ribosomal peptides (NRPS)], orange (T1PKS), light blue (NRPS+T1PKS hybrids), purple (terpenes), and red (siderophores). Links connect homologous regions; single nucleotide polymorphisms (SNPs) are highlighted in black. (B) Stacked bar graph comparing the abundance of BGCs between strains. CFAM 0234 shows greater diversity, with unique clusters (*e.g.*, betalactones).

TABLE III

Synthesis of relevant classes of biosynthetic gene clusters (BGCs) and gene clusters families (GCFs) in Aspergillus japonicus

| BGC class | Strain | Distribution (%) | Relevant GCFs | Features/Potential |
|---|---|---|---|---|
| NRPS | CFAM 0234 | 45% (48/106) | FAM_00002 (Shared) | Synthesis of non-ribosomal peptides |
| | CBS 114.51 | 38% (22/57) | FAM_00125 (CBS Exclusive) | Unusual enzyme domains |
| T1PKS | CFAM 0234 | 30% (32/106) | FAM_00001 (Shared) | Biosynthesis of polyketides (e.g. pigments) |
| | CBS 114.51 | 35% (20/57) | FAM_00096 (CFAM Exclusive) | T1PKS + Isocyanide hybrid cluster |
| Terpenes | CFAM 0234 | 12% (13/106) | FAM_00057 (CFAM Exclusive) | Production of uncharacterized terpenoids |
| | CBS 114.51 | 15% (9/57) | FAM_00125 (CBS Exclusive) | Atypical modular architecture |
| Hybrids (NRPS+T1PKS) | CFAM 0234 | 8% (9/106) | FAM_00009 (Shared) | Antifungal hybrid metabolites |
| | CBS 114.51 | 7% (4/57) | FAM_00125 (CBS Exclusive) | Rare addition domains (ex.: epoxidase) |
| Betalactones | CFAM 0234 | 4 clusters (Exclusive) | FAM_00024 (Shared) | Potential for β-lactam antibiotics |
| Indole/Other | CFAM 0234 | 5% (5/106) | FAM_00010 (CBS Exclusive) | Unmapped secondary roads |
| | CBS 114.51 | 5% (3/57) | FAM_00055 (CFAM Exclusive) | Absent in databases (evolutionary novelty) |

TABLE IV

Determination of antibacterial activity using the agar diffusion technique per well

| Extract | Bacteria tested (halo mm) | | | | | | | | | | | | | |
|---|---|---|---|---|---|---|---|---|---|---|---|---|---|---|
| | *A. n* 0725 | *E. f* 0639 | *E. f* 0735 | *E. c* 0001 | *E. c* 0002 | *K. a* 0717 | *K. p* 0462 | *P. m* 0686 | *P. a* 0024 | *S. e* 0047 | *S. t* 0009 | *S. f* 0046 | *S. s* 0036 | MRSA 4330 |
| *Aspergillus japonicas* 0234 | - | - | - | 8 | - | - | - | - | - | - | - | - | 8 | 6 |

*A. n*: *Acinetobacter nosocomialis*; *E. f*: *Enteroccocus faecium*; *E. c*: *Escherichia coli*; *K. a*: *Klebsiella aerogenes*; *K. p*: *Klebsiella pneumoniae*; *P. m*: *Proteus mirabilis*; *P. a*: *Pseudomonas aeruginosa*; *S. e*: *Salmonella entérica*; *S. t*: *Salmonella typhi*; *S. f*: *Shigella flexneri*; *S. s*: *Shigella sonnei*; MRSA: *Sthapylococcus aureus*.

## DISCUSSION

The production of secondary metabolites is strongly influenced by culture conditions, and the present work was focused on exploring the genomic potential of *A. japonicus* CFAM 0234. The antimicrobial activity observed with the crude extract supports the hypothesis that some predicted biosynthetic clusters are active. Future work will aim to validate these predictions through metabolomic analyses under different growth conditions and extraction methodologies.

The relevance of the antimicrobial activity of *A. japonicus* CFAM 0234 against *E. coli*, *S. sonnei* and *S. aureus* (MRSA). Inhibition zones between 6 - 10 mm were considered indicative of moderate activity when compared with the reference antibiotic control. Although the agar disk diffusion assay is qualitative and does not allow precise quantification of the antimicrobial agent,[19] it provides preliminary evidence of bioactivity. Determination of the minimum inhibitory concentration (MIC) will be necessary in future work to accurately quantify antimicrobial potency.

Recently, the World Health Organisation[20] classified these bacteria among the priority pathogens for research into new antibiotics, reinforcing the importance of exploring therapeutic alternatives. A study by Naddaf[21] reveals that deaths linked to resistant gram-negative bacteria, such as *E. coli*, have increased by almost 150%, from 50,900 cases in 1990 to 127,000 cases in 2021.

Shigellosis, which causes around 700,000 deaths annually — mainly in children in low- and middle-income countries[22] — and the prevalence of *S. sonnei* in outbreaks in Brazil,[23] emphasise the urgent need for new approaches.

According to data reported to Agência Nacional de Vigilância Sanitária (ANVISA) in 2019, MRSA accounted for 59.1% of *S. aureus* samples isolated from hospitalised adult patients and was one of the most lethal pathogen-drug combinations, resulting in approximately 121,000 deaths attributable to antimicrobial resistance, according to a study by the Institute for Health Metrics and Evaluation.[24,25,26,27]

The experimental results show that the ethyl acetate extract of CFAM 0234 inhibits *E. coli*, *S. sonnei* and *S. aureus* (MRSA), demonstrating antimicrobial activity. This finding is corroborated by the literature which documents the resistance of these pathogens to antibiotics such as amoxicillin and cephalothin,[28,29] indicating that the metabolites produced by CFAM0234 can act by alternative mechanisms and bypass the resistance mechanisms.

Genomic analysis revealed important information that corroborates the phenotypic findings. The genome of CFAM 0234 was assembled into 962 contigs (total size of 72.67 Mbp, N50 = 152 kbp), although the assembly is fragmented - a fact also seen in the reference strain CBS 114.51 (36.1 Mbp, N50 = 412 kbp). The comparison between the two strains, carried out using OrthoFind-

er, showed significant differences in the repertoire of genes and suggested the presence of specific genomic regions with functional potential, as observed in specific contigs (*e.g.*, Contig_591, Contig_606 and Contig_681).

In terms of biosynthetic potential, CFAM 0234 presented a significantly expanded repertoire, with 106 BGCs identified compared to 57 in strain CBS 114.51. Analysis with BiG-SCAPE revealed that 63 BGC families (GCFs) are unique to CFAM 0234, while 33 are unique to the reference, demonstrating a disparity that may reflect evolutionary adaptations to Amazonian soil. The comparative map (Figure A) shows the association between structural rearrangements and the location of BGCs — especially those of NRPS and T1PKS —, with a high density of SNPs indicating selective pressure for metabolic diversification.[30]

Additionally, the correlation between the antimicrobial activity of the crude extract and the detection of 14 BGCs, including five NRPS clusters with homology to gliotoxin-producing systems, two type I PKS clusters similar to those involved in lovastatin synthesis and a hybrid NRPS-PKS cluster containing chloramphenicol acetyltransferase domains, reinforces the potential of novel compounds. Although the activity of the extract is moderate (8 mm halo), this evidence motivates the application of purification techniques, such as liquid chromatography, to isolate the active compounds.

In addition, the abundance and diversity of BGCs, especially the NRPS, PKS and hybrid clusters, suggest promising biotechnological potential. While NRPS clusters are historically related to the synthesis of non-ribosomal peptides (*e.g.*, penicillin), NRPS-PKS hybrid systems can produce molecules with unique structures, such as rapamycin, already recognised for its immunosuppressive properties.[31] The identification of unique clusters, such as those of betalactones — potential precursors of β-lactam antibiotics — further highlights the possibility of discovering new bioactive compounds.[32]

Integrating the genomic data with the results of the biological tests strengthens the hypothesis that the antimicrobial activity observed is not fortuitous, but rather the result of the expression of secondary metabolites derived from the BGCs identified. The concentration of SNPs in regulatory regions of BGCs (such as NRPS promoters) suggests variations in gene regulation between strains, possibly influenced by specific environmental pressures.[33,34] In addition, structural rearrangements near PKS clusters may be promoting genetic recombination that diversifies metabolic products, an evolutionary strategy already demonstrated in fungal studies.[35]

Finally, the fragmentation of the genome (N50 = 152 kbp) and the lack of correspondence with the MIBiG database highlight limitations that can be overcome with the use of long-read sequencing technologies (Oxford Nanopore) and metabolomic profiling techniques [liquid chromatography-mass espectrometry (LC-MS)]. These approaches would be fundamental for the functional validation of the predicted BGCs and for the isolation of the metabolites responsible for antimicrobial activity, paving the way for the development of new drugs.

*In conclusion* - This study has shown that *A. japonicus* CFAM 0234 has a broad and diverse repertoire of biosynthetic clusters, which is reflected in its antimicrobial activity against *E. coli*, *S. sonnei* and *S. aureus*. The genomic and phenotypic data indicate that the secondary metabolites produced by this strain may offer promising alternatives to combat resistant pathogens, in line with the priorities established by the WHO. Although the fragmentation of the genome and the lack of correspondence with the MIBiG bank represent methodological limitations, these issues open up avenues for the application of advanced techniques, such as long-read sequencing and integration with metabolomic analyses, in order to validate and fully exploit the biotechnological potential identified.

Therefore, the results presented not only contribute to the understanding of the resistance mechanisms of these bacteria and the bioactive factors in Amazonian fungi, but also establish a solid basis for future investigations that could lead to the development of new antimicrobial agents. The continuation of this work, with the improvement of analytical approaches and the functional validation of the predicted clusters, promises to significantly advance the discovery of innovative compounds to combat emerging infections.

## ACKNOWLEDGEMENTS

To Amazon Bacteria Collection - CBAM and Amazon Fungi Collection - CFAM

## AUTHORS' CONTRIBUTION

JCR - participation in the antimicrobial activity test against the clinical test microorganisms. Furthermore, analysed the results obtained for data dissemination and participated in the writing of the manuscript; MEG - performance of the bioinformatics analyses, analysed the results obtained for data dissemination and participated in the writing of the manuscript; ASQ and ALRL - participation in the antimicrobial activity test against the clinical test microorganisms; CVSG - reactivation of the selected fungal culture for the performance of the activities and also in the writing of the manuscript in the English language; LML - collaboration with the involved researchers to ensure that the research was conducted with quality. Additionally, they analysed the results obtained for data dissemination; OCCF - collaboration with the involved researchers to ensure that the research was conducted with quality. The authors declare no conflicts of interest.

## DATA AVAILABILITY

Raw genomic data were deposited in the NIH's genetic sequence database - GenBank.

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

# OPEN PEER REVIEW

Memórias do IOC thanks the anonymous reviewers for their contribution to the peer review of this work.

## FIRST REVIEW ROUND

### REVIEWERS' COMMENTS

**REVIEWER #1**

The article is interesting; however, I believe that, to make it more compelling, the authors should demonstrate that the substances identified in the genome are indeed being produced by the fungus. To achieve this, different methodologies for secondary metabolite production should be employed, including variations in culture media and extraction methods, in order to determine the best approach for producing the metabolites of interest.

1) The culture medium used for fungal growth for DNA and secondary metabolite extraction should be the same, since, depending on the culture medium used, different metabolites may be produced. Therefore, the fungus grown in YES medium may not produce the same metabolites that are encoded in the genome.

2) What would be considered moderate antimicrobial activity, as you mentioned in the discussion and conclusion(line 258 and 310)? What parameter was used for this classification?
The disk diffusion method for extracts and non-standardized substances is only a qualitative method, and therefore it is not possible to quantify the amount of antimicrobial activity. I suggest performing the minimum inhibitory concentration (MIC) test to determine more precisely whether the antimicrobial activity of the extract is noteworthy.

3) I missed having a table containing the disk diffusion test results with all the bacteria tested.

### AUTHORS' RESPONSE TO THE REVIEWERS

Question 1) The culture medium used for fungal growth for DNA and secondary metabolite extraction should be the same, since, depending on the culture medium used, different metabolites may be produced. Therefore, the fungus grown in YES medium may not produce the same metabolites that are encoded in the genome.

Response: We thank the reviewer for this important observation. We fully agree that the expression of biosynthetic gene clusters (BGCs) and the production of secondary metabolites are highly dependent on environmental and nutritional factors. The main purpose of the present study was to explore the genomic potential of Aspergillus japonicus CFAM 0234, identifying biosynthetic clusters with biotechnological relevance through comparative genome mining.

Although we did not perform metabolomic validation under different culture conditions, the detection of selective antimicrobial activity in the crude extract provides preliminary experimental evidence that some biosynthetic pathways may indeed be active. We have clarified this in the revised Discussion (lines 240 - 245) and explicitly stated that further experiments involving optimization of culture media, extraction protocols and LC-MS metabolomic profiling will be carried out to link BGCs to specific metabolites.

Question 2) What would be considered moderate antimicrobial activity, as you mentioned in the discussion and conclusion (line 258 and 310)? What parameter was used for this classification?

The disk diffusion method for extracts and non-standardized substances is only a qualitative method, and therefore it is not possible to quantify the amount of antimicrobial activity. I suggest performing the minimum inhibitory concentration (MIC) test to determine more precisely whether the antimicrobial activity of the extract is noteworthy.

Response: We thank the reviewer for pointing this out. We have now clarified in the Materials and Methods and Discussion sections that the classification of "moderate activity" was based on halo diameters between 6 and 10 mm, following comparative evaluation with positive (vancomycin, 1 mg/mL) and negative controls (DMSO 4%).

We fully agree that the disk diffusion test is a qualitative approach and that it does not allow precise quantification of antimicrobial activity, as the amount of compound diffused into the agar cannot be accurately determined. As highlighted in the literature (Balouiri et al, 2015), disk diffusion is widely used for preliminary antimicrobial screening of extracts and other compounds due to its simplicity, low cost, and ability to test multiple microorganisms simultaneously, but MIC determination is required for accurate assessment of potency. We have added a note acknowledging this limitation and emphasizing that MIC and purification assays will be addressed in future studies. (changes in line 247-253)

Question 3) I missed having a table containing the disk diffusion test results with all the bacteria tested.

Response: We appreciate the suggestion. The revised manuscript now includes a summary table (Table 4) listing the inhibition halo diameters for all tested bacterial strains, as obtained in the disk diffusion assays. This addition allows a clearer overview of the selectivity profile of the crude extract.

## SECOND REVIEW ROUND

**REVIEWERS' COMMENTS**

**REVIEWER #1**

No comments.

