## [Reviewer Report · FIRST REVIEW ROUND - REVIEWERS COMMENTS]

## REVIEWER #1

The article is interesting; however, I believe that, to make it more compelling, the authors should demonstrate that the substances identified in the genome are indeed being produced by the fungus. To achieve this, different methodologies for secondary metabolite production should be employed, including variations in culture media and extraction methods, in order to determine the best approach for producing the metabolites of interest.

1) The culture medium used for fungal growth for DNA and secondary metabolite extraction should be the same, since, depending on the culture medium used, different metabolites may be produced. Therefore, the fungus grown in YES medium may not produce the same metabolites that are encoded in the genome.

2) What would be considered moderate antimicrobial activity, as you mentioned in the discussion and conclusion (line 258 and 310)? What parameter was used for this classification? The disk diffusion method for extracts and non-standardized substances is only a qualitative method, and therefore it is not possible to quantify the amount of antimicrobial activity. I suggest performing the minimum inhibitory concentration (MIC) test to determine more precisely whether the antimicrobial activity of the extract is noteworthy.

3) I missed having a table containing the disk diffusion test results with all the bacteria tested.

## AUTHORS' RESPONSE TO THE REVIEWERS

**Question 1)** The culture medium used for fungal growth for DNA and secondary metabolite extraction should be the same, since, depending on the culture medium used, different metabolites may be produced. Therefore, the fungus grown in YES medium may not produce the same metabolites that are encoded in the genome.

*Response:* We thank the reviewer for this important observation. We fully agree that the expression of biosynthetic gene clusters (BGCs) and the production of secondary metabolites are highly dependent on environmental and nutritional factors. The main purpose of the present study was to explore the genomic potential of *Aspergillus japonicus* CFAM 0234, identifying biosynthetic clusters with biotechnological relevance through comparative genome mining.

Although we did not perform metabolomic validation under different culture conditions, the detection of selective antimicrobial activity in the crude extract provides preliminary experimental evidence that some biosynthetic pathways may indeed be active. We have clarified this in the revised Discussion (lines 240-245) and explicitly stated that further experiments involving optimization of culture media, extraction protocols and LC-MS metabolomic profiling will be carried out to link BGCs to specific metabolites.

**Question 2)** What would be considered moderate antimicrobial activity, as you mentioned in the discussion and conclusion (line 258 and 310)? What parameter was used for this classification? The disk diffusion method for extracts and non-standardized substances is only a qualitative method, and therefore it is not possible to quantify the amount of antimicrobial activity. I suggest performing the minimum inhibitory concentration (MIC) test to determine more precisely whether the antimicrobial activity of the extract is noteworthy.

*Response:* We thank the reviewer for pointing this out. We have now clarified in the Materials and Methods and Discussion sections that the classification of "moderate activity" was based on halo diameters between 6 and 10 mm, following comparative evaluation with positive (vancomycin, 1 mg/mL) and negative controls (DMSO 4%).

We fully agree that the disk diffusion test is a qualitative approach and that it does not allow precise quantification of antimicrobial activity, as the amount of compound diffused into the agar cannot be accurately determined. As highlighted in the literature (Balouiri et al, 2015), disk diffusion is widely used for preliminary antimicrobial screening of extracts and other compounds due to its simplicity, low cost, and ability to test multiple microorganisms simultaneously, but MIC determination is required for accurate assessment of potency. We have added a note acknowledging this limitation and emphasizing that MIC and purification assays will be addressed in future studies. (changes in line 247-253)

**Question 3)** I missed having a table containing the disk diffusion test results with all the bacteria tested.

*Response:* We appreciate the suggestion. The revised manuscript now includes a summary table (Table 4) listing the inhibition halo diameters for all tested bacterial strains, as obtained in the disk diffusion assays. This addition allows a clearer overview of the selectivity profile of the crude extract.